# *"Younger women had more access to COVID-19 information"*: An intersectional analysis of factors influencing women and girls' access to COVID-19 information in Rohingya and host communities in Bangladesh

**Ateeb Ahmad Parray** [ID]*, **Muhammad Riaz Hossain** [ID], **Rafia Sultana** [ID], **Bachera Aktar** [ID], **Sabina Faiz Rashid** [ID]

The Center of Excellence for Gender, Sexual and Reproductive Health and Rights, BRAC James P Grant School of Public Health, BRAC University, Dhaka, Bangladesh

* ahmad.ateeb101@gmail.com

**Data Availability Statement:** All relevant data are within the paper.

## Abstract

The Rohingya and Bangladeshi host communities live at a heightened risk of COVID-19 impact due to their pre-existing vulnerabilities, religious beliefs, and strict socio-cultural and gender norms that render primarily women and girls vulnerable. However, the extent of this vulnerability varies within and across population groups in the host and Rohingya communities. The intersectionality lens helps identify, recognize, and understand these factors that create inequities within populations. This study explored the factors that influenced the women and girls' access to information during the COVID-19 pandemic through an intersectional lens. This paper presents partial findings from the exploratory qualitative part of mixed-method research conducted in ten Rohingya camps and four wards of the adjacent host communities in Cox's Bazar, Bangladesh. Data were extracted from 24 in-depth interviews (12 in each community) conducted from November 2020 to March 2021 with diverse participants, including adolescent girls, younger women, adult women, pregnant and lactating mothers, persons with disabilities, older adults, and single female-household heads. All participants provided verbal informed consent before the interviews. In the case of the adolescents, assent was taken from the participants, and verbal informed consent was taken from their parents/guardians. The ethical clearance of this study was sought from the institutional review board of BRAC James P Grant School of Public Health, BRAC University. We find that the women and girls living in Rohingya communities exhibit a more profound structural interplay of factors within their socio-ecological ecosystem depending on their age, power, and position in the society, physical (dis)abilities, and pre-existing vulnerabilities stemming from their exodus, making them more vulnerable to COVID-19 impact by hindering their access to information. Unlike Rohingya, the host women and girls explain the impact of the COVID-19 pandemic on their access to information through the lens of intergenerational poverty and continuous strain on existing resources, thereby highlighting shrinking opportunities due to the influx, COVID-19 infodemic and misinformation, access to

**Funding:** The project that this study is based on, was funded by International Development Research Centre, Canada under Grant No. 109493 – 001. The funder had no role in the design and conduct of the study; collection, management, analysis, and interpretation of the data; preparation, review, or approval of the manuscript; and decision to submit the manuscript for publication.

**Competing interests:** The authors have declared that no competing interests exist.

digital devices amongst the adolescents, and restricted mobility mainly due to transport, school closures, and distance-related issues. Moreover, the socio-cultural beliefs and the gender norms imposed on women and adolescent girls played an essential role in accessing information regarding the COVID-19 pandemic and consequently influenced their perception of and response to the disease and its safety protocols. Socio-cultural gender norms led to mobility restrictions, which compounded by lockdowns influenced their access to information resulting in dependency on secondary sources, usually from male members of their families, which can easily mislead/provide mis/partial information. The younger age groups had more access to primary sources of information and a broader support network. In comparison, the older age groups were more dependent on secondary sources, and their social networks were limited to their family members due to their movement difficulty because of age/aging-related physical conditions. This study explored and analyzed the intersectional factors that influenced the women and girls' access to information during the COVID-19 pandemic from two contexts with varying degrees of pre-existing vulnerability and its extent. These include gender, age, state of vulnerability, power and privilege, socio-economic status, and physical (dis)ability. It is imperative that services geared towards the most vulnerable are contextualized and consider the intersectional factors that determine the communities' access to information.

## Introduction

Intersectionality stems from the work of African American feminist scholars [1–3]. It adopts multiple categories of analysis, such as race, gender, sex, ethnicity, socio-economic status, class, marital status, etc., to analyze the various tenets of social identities and the interactions between them [4]. It considers what occurs when multiple axes of inequalities enter each other in a particular context and ascertain that focusing on only one axis of inequality at a time hinders the natural pandemic effect and its unequal distribution across social groups within the society [5, 6]. Therefore, the application of intersectionality offers to learn the interplay between different factors and helps in the identification of deep-rooted vulnerabilities and advantages to uncover the complex social influencing factors and power structures that create, sustain, and often intensify them [7].

The intersectionality lens is increasingly being adopted in global health research to understand how multiple dimensions of one's identity can intersect and add to social inequalities and health disparities within and across subgroups of most-vulnerable populations [8]. For instance, the social and health inequities that limit access to information for such groups are compounded by intersectional factors such as gender, age, socio-economic status, pre-existing vulnerabilities, power, privilege, physical disability, and religion [5–9]. In any humanitarian crisis (e.g., the Rohingya Refugee crisis), communities live in an already vulnerable state compounded by the COVID-19 pandemic [10, 11]. Communities who have fled violence, genocide, persecution, natural disasters, etc., and are living in confined camps are likely to have less access to information than those in the host communities (local populations). Conversely, communities in the camps may have greater access to resources as more humanitarian workers may concentrate their services in the centers. In such a situation, many intersectional factors influence how people access information about/regarding the pandemic. The intersectionality lens helps identify, recognize, and understand these factors that create such inequities.

Since August 2017, more than one million Rohingya have been residing in Cox's Bazar, Bangladesh [12]. These refugees live in the world's most densely populated 34 refugee camps in Ukhiya and Teknaf sub-districts in the Cox's Bazar [11], one of the poverty-stricken southern districts of Bangladesh. The Rohingya have limited freedom of movement outside the camps, reduced access to necessities, and live in vulnerable conditions heightened by their history of fleeing a genocide [13], making them especially vulnerable to COVID-19 impact. There is compelling evidence that the COVID-19 pandemic is inflicting varying degrees of negative implications on the refugee camps and local communities, including effects on social security, safety, social cohesion, shrinking livelihood opportunities, and erosion of dignity and well-being [14–18], particularly for Most Vulnerable groups (MVGs)—Single women who are household heads, pregnant and lactating mothers, adolescents, youth, elderly, and persons with disabilities [19]. We define "*Vulnerabilities*" as the characteristics of an individual, household, or sub-group within a given population and their situation in Cox's Bazar that influence their capacity to meet basic needs, as well as their exposure to physical or mental harm [2].

Due to their restricted mobility in the camps, the MVGs exhibit minimal education and skill development [20]. They are more exposed to physical and mental harm due to their gender roles and caregiving responsibilities, which also render them dependent on other individuals and have been compounded by the COVID-19 pandemic, making them highly vulnerable to abuse, intimidation, sexual and gender-based violence [20]. The Rohingya are a conservative community with rigid social and cultural norms that frown upon and create undue tensions around women and girl empowerment. In general, Rohingya Women and girls experience barriers to mobility and access to and control over resources [21].

Hindsight hints that the increased domestic violence in the home and harassment outside it for Rohingya women and girls may be attributed to the increased paid work for women [21]. A Rapid Gender Analysis (RGA) conducted with Rohingya reported that every woman and girl was either a survivor of sexual assault or a witness to it from their time in Myanmar, but the same woman or girl felt relatively safe in camps in Bangladesh [22]. However, as stateless people dependent on relief and humanitarian aid, all Rohingya in camps in Bangladesh are vulnerable. Various reports have shown that COVID-19 pandemic-induced lockdowns, cramped conditions, lack of appropriate sanitation facilities, and disrupted services put women and girls at risk of gender-based violence (GBV), including sexual harassment and assault, and sexual violence [20].

Women's mobility is also hindered by the observance of purdah—a religious doctrine imposed on women and girls that requires them not to be seen by any male person except their family members (*Mahrams*), which limits their ability to access aid, livelihood opportunities, humanitarian services including GBV services [22]. Adolescent girls are generally more vulnerable to GBV threats and have very restricted mobility outside the home, so their access to services and information is even more limited [17]. However, many of them have access to digital media (mobile phones, social media, messaging applications, etc.) and tend to be comparatively more updated than the others [23]. Compounded by the onset of COVID-19 and strict lockdowns restricting all essential and non-essential movements in the camps, women, and girls had to suffer the greater brunt of the COVID-19 crisis [23].

The Rohingya are also surrounded by around half-million Bangladeshi host communities—one of the poorest populations in Bangladesh with a poverty rate of 33% [7]. In August 2017, the host communities gathered resources of many forms and quantities to alleviate the refugees' sufferings, even before the Bangladesh government committed to the humanitarian effort formally [24]. However, within a year of the previous refugee arrival in 2017, visible solidarity and support for the migrants waned. As COVID-19 approached, the hostility against refugees and humanitarian assistance organizations was palpable, especially among the impoverished

local people in Cox's Bazar region [25, 26]. The early negative implications of refugee settlement were the most severe for the hosts due to the scarcity of resources and possibilities. Additionally, the degradation of public services such as birth and death registration fuelled dissatisfaction against the government's overall approach to the refugee crisis [24]. Inflation in the pricing of critical goods, declining salaries for low-skilled employees, competition for scarce natural resources, and loss of livelihood exacerbated the tension between refugees and Bangladeshi host communities [27]. In general, the host community believes that the government of Bangladesh's unilateral humanitarian assistance to refugees demonstrates a failure of the government to recognize the harmful effect of poverty on the indigenous population in a number of its policies and initiatives [18].

Caused by overpopulation and a lack of privacy, women in host communities have claimed increasing restrictions on their freedom of movement and voiced fear of the Rohingya [27]. In the host community, the risk of GBV is significant and is expected to rise during times of economic hardship [10, 24]. As is the case with the Rohingya, child marriage is prevalent in the host society and is employed as a coping mechanism by the poorest and most vulnerable families during times of crisis [22]. Domestic violence, mainly IPV is also common in both communities [28]; women are the primary victims, while their husbands are the abusers, with an increased risk of domestic violence in the Rohingya community since their relocation [22]. This is considered an internal family matter requiring no external intervention. Female-headed families, particularly those with no male representation, are significantly more susceptible [28]. According to one study conducted by ACAPS with the host community, 45 percent of female-headed families in the host community were vulnerable or very vulnerable in December 2017, compared to 35 percent of male-headed households [28]. Another research, based on a December 2017 evaluation, suggests that food insecurity is almost as severe for women in the host community as it is for Rohingya women, with just one in three women having access to a diverse diet [29]. Female-headed households, in particular, have been identified to be among the most vulnerable [11, 30]. However, this list omits single women who traveled alone during the exodus and their difficulties in providing for basic requirements—traveling outside the house to gather daily essentials, firewood, or water, seeking healthcare for themselves and their children, and generating an income. Due to a shortage of income-generating options for women, they and their families are financially insecure in host communities [28, 31].

The COVID-19 pandemic has also exacerbated the current fragility of living conditions, healthcare, disaster management (floods, fires, etc.), other services and facilities, and increasing competition over scarce resources between the refugee and host communities [32]. Due to the COVID-19 pandemic, the women and girls, including the persons with disabilities and the elderly in these communities faced added barriers to accessing necessities such as healthcare consultations or freedom of movement outside the house due to their society's deep-rooted existing gendered, social, cultural, and religious norms [8]. Evidence suggests that adult women and children among Rohingya, are most vulnerable to the COVID-19 due to their misconceptions and lack of awareness regarding the same [14]. In contrast, older adults are more susceptible to COVID-19 due to age-related complications [33, 34]. These factors play an important role in how different groups, especially the most vulnerable within the two communities received information about and responded to the COVID-19 pandemic.

Despite the issue's urgency and effect on the host community, there has been a shortage of pertinent research addressing the complicated nexus between the refugee crisis and host communities in Bangladesh, as well as the impact of COVID-19 on both populations in terms of access to COVID-19 information [35]. Recent work has concentrated on poverty, livelihoods, security issues, illicit trafficking, and environmental degradation but has largely overlooked their access to COVID-19 information and services [28, 29, 32]. By bringing host communities

and the existing COVID-19 crisis and its implications into the center of the analysis, the paper underlines the need for a more nuanced rather intersectional approach involving the MVGs in both communities and putting them at the locus of the humanitarian response. We analyze the interplay of intersectional factors that influenced the access to COVID-19 information of women and girls in the Rohingya camps and host communities in Cox's Bazar.

## Materials and methods

This study was part of a larger participatory action research project [36] that explored the most-vulnerable groups (MVGs) and their perceptions, needs and impact during the COVID-19 pandemic. An exhaustive literature review was undertaken to list the MVGs from the Rohingya and host communities. These lists were further reinforced during the expert consultation workshops done with humanitarian researchers and practitioners working in the Rohingya Humanitarian Response, Cox's Bazar, in November 2020 to identify the MVGs and the factors that determine their vulnerabilities. The final list of MVGs was developed after validation through field visits and face-t0-face interviews with a diverse range of people from both communities. Since this paper focuses on women and girls, we have thus included a sub-set of the data collected regarding MVGs in both the Rohingya and host contexts, including pregnant and lactating mothers, female persons with disabilities (PWDs), adolescent girls, single female household heads (SFHH) and the female older adults (elderly), thereby excluding the male MVGs, i.e., adolescent boys, young male adults, and the male elderly.

This qualitative study was conducted in 10 selected Rohingya refugee camps in the Ukhiya sub-district of Cox's Bazar, Bangladesh, and four adjoining wards of the Rajapalong union in the surrounding host communities. The data collection took place from November 2020 to March 2021. We employed an exploratory qualitative design whereby 24 in-depth case studies, 12 in each Rohingya and the host community, were conducted with purposively selected participants from the list of MVGs identified in the larger research project, serving as inclusion criteria for recruitment to this study.

An in-depth interview (IDI) guideline was developed to conduct the case studies and was pre-tested in both the Rohingya and host communities. Some modifications were made to the policy after the pilot. Because of the language barrier, the researchers paired with locally recruited trained data enumerators fluent in both the local Bangla (the host community's language) and Rohingya dialects (Ruáingga) and acted as translators while conducting qualitative interviews. The data was collected in both Bangla and Ruáingga. Each interview lasted for an average of 85 minutes. After data collection, the recorded data were transcribed verbatim in Bangla within 24 hours by the local enumerators and then translated into English by a group of professional translators. After that, the researchers cross-checked the data and familiarized themselves with it. Subsequently, inductive coding was done to identify key themes and sub-themes. Thematic analysis was conducted using the qualitative software—Atlas Ti, version 9.0.

### Ethics statement

The Institutional Review Board of BRAC James P Grant School of Public Health, BRAC University provided the ethical clearance for the study under reference number IRB-6 November 20–057. The clearance for the broader research project was sought and obtained from the Refugee, Relief, and Repatriation Commissioner (RRRC), the Camp-in-Charges (CiC) of all the selected camps, the Union Parishad Office, and local community leaders from both communities. Verbal informed consents were obtained from all participants. Informed verbal Assent was also taken from the adolescent participants after consent was taken either from their guardian(s) or parent(s). Participant names were replaced with pseudonyms, and only limited

personal information important for analysis was collected, such as age, gender, occupation, and education. Data, including Bangla and English transcripts, were stored in a secured locker at the home institute of the researchers, where only the selected research team members had access. Furthermore, because this research was conducted during a pandemic, the research team followed recommended COVID-19 safety precautions during fieldwork to prevent coronavirus transmission to community members, research participants, and researchers. All the IDI and KII participants were given hygiene packets (face masks, soap, toothpaste, toothbrush shampoo) as a token of appreciation for their participation in the study.

## Results

### Respondents profile

Among the 24 participants, 12 were from the Rohingya community, and 12 were from the host community: participants included adolescent girls, married and unmarried women, pregnant women, single female household heads, persons with disabilities (PWD), and older adults. In the host community sample, all adult women were involved in income generation activities, such as community volunteers, small business owners, and private jobholders. However, in the Rohingya sample, except for one participant, all were homemakers. One-third of the host participants identified themselves as Hindus while all the Rohingya participants identified themselves as Muslims. Table 1 presents the profile of the respondents included in this study.

### Intersectional factors influencing women and girls' access to COVID-19 information

Our analysis found that the intersectional factors that influenced the women and girls' access to information during the COVID-19 pandemic include gender, age, socio-economic status, state of vulnerability, power, and privilege, and physical (dis)ability (Fig 1).

**Table 1. Profile of the respondents included in this study.**

| Characteristic | Group | |
| --- | --- | --- |
| **Type of Participants** | **Rohingya participants** | **Host community participants** |
| Single female household heads | 3 | 3 |
| Persons with disabilities | 2 | 2 |
| Adolescents | 2 | 2 |
| Pregnant and Lactating Women | 3 | 3 |
| Elderly | 2 | 2 |
| **Total (N)** | **12** | **12** |
| **Age (In Years)** | | |
| 11–17 | 2 | 2 |
| 18–64 year | 8 | 8 |
| Above 65 year | 2 | 2 |
| **Occupation** | | |
| Homemakers | 10 | 8 |
| Community volunteers | 1 | 1 |
| Small business | - | 1 |
| Student | - | 1 |
| Private job | 1 | 1 |
| **Religion** | | |
| Muslim | 12 | 8 |
| Hindu | - | 4 |

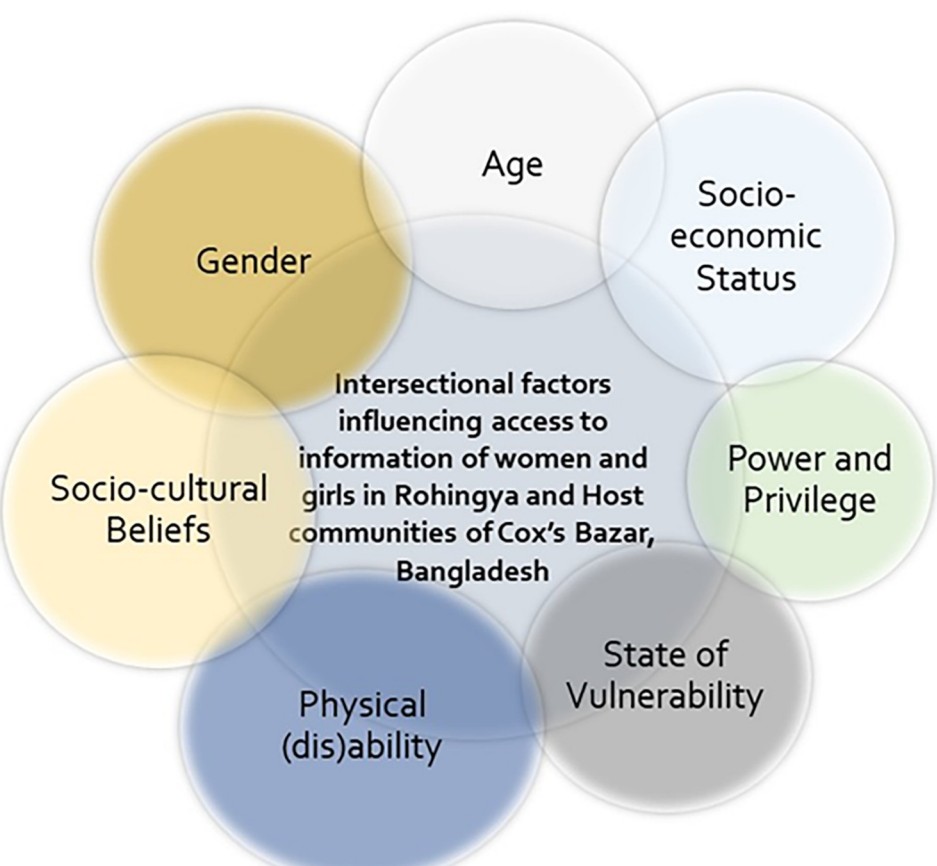

**Fig 1. Intersectional factors influencing women and girls' access to information during COVID-19 pandemic in Rohingya and Host communities of Cox's Bazar.**

## Socio-cultural beliefs and gender norms limit access to accurate information

Our data showed that socio-cultural beliefs and gender norms imposed on women and adolescent girls played an important role in how they accessed information regarding the COVID-19 pandemic and consequently influenced their perception of and response to the disease.

Rooted in orthodox religious values, the socio-cultural beliefs in both the Rohingya and host communities restrict their mobility to public spaces and thus, limit their access to information beyond their immediate homes. As a result, they usually receive information from secondary sources such as the male family members resulting in misinformation being spread within households, especially among women. Often important pieces of information are changed or left out in the process of being relayed. Although many participants referred to community health workers (CHWs) visiting door-to-door and disseminating COVID-19 awareness information, however, in many instances, participants were unable to talk to CHWs because of their household chores or did not have permission from their husbands or male household heads (for example, fathers/fathers-in-law/brother) for interacting with any outsider (including male CHWs) and therefore lacked primary sources of information.

For instance, an adult Rohingya female participant informed that she first heard about COVID-19 (that it was a disease and had originated in China) in April 2020 from her husband,

who had heard about it from other men of their community at the mosque after prayers. Since she did not venture out of the home, her only source of information was her husband. The secondary sources of information often lead to the relay of partial information or incorrect information within the family members. For instance, an elderly Rohingya woman shared that since she was sick and lived with her son, she rarely ventured outside her house. Her son had told her that COVID-19 was a "flu". As such, she could not understand how it could be a deadly disease, *"My son says it is a flu...If it's just like the flu then it will go away also... Why be so afraid?"*, she mentioned.

Rohingya adolescent girl participants informed that culturally, unmarried Rohingya adolescent girls are not permitted to go outside their houses and their male family members bring them their daily necessities. Therefore, they most often do not have exposure to the outside world including any sources of information other than their family members. As one 16-year-old Rohingya adolescent girl said that she did not know about COVID-19 because she did not go outside her house, and no one shared anything about this. She mentioned that she practiced "*purdah*"–a religious doctrine that required her to be seen by no male persons except for her family members. Such strict practices invigorated altruistic notions about COVID-19 among a few participants. For instance, one 26-year-old woman with a disability mentioned that's he cared less about COVID-19 as it had come from *Allah* and she practiced *purdah* which means that even if she died due to COVID-19, she would go to the *Jannat* (heaven). When asked where she heard about COVID-19 from and if she had spoken to any CHW before, she mentioned,

> *"My husband told me about COVID-19...He is a god-loving person...He said that Kiyyamat (doomsday) is near and I should practice purdah even more strictly"*.

This quote highlights the extent of trust and dependency Rohingyas women exhibit in their male counterparts who due to rigid socio-cultural beliefs, often act as barriers to their access to COVID-19 information or/and relay incorrect or partial information. This, compounded by the participant's physical disability and the COVID-19 lockdown put these women and girls in hostile and supportless situations in the camps.

The situation was similar in the host community, albeit the socio-cultural norms there were somewhat lenient than in the Rohingya community. As such, some women were able to access information on their own and relay it to their peers, neighbors, and family members. For instance, one young lactating mother in the host community said that she had first heard about COVID-19 from her friend who worked as a volunteer for a local NGO. Even though, because of her small child, she was confined to her home, her friend would come to her once a week and talk to her about the pandemic. As she said,

> "*I can't go outside because my baby is small, and my in-laws don't let me go out... but my friend can work. So, when she gets any new information, she comes and tells me.*" (Lactating Mother, Host)

In the host community, adolescent girls had greater access to primary sources of information as they were able to talk to their peers and access social media through their mobile devices or the mobile devices of their family members. However, some adolescent girls mentioned that due to the closure of educational institutions they were unable to talk to their peers as they were barred from going outside their homes. The elderly women in the host community had similar experiences to those in the Rohingya community and were mostly unaware of the severity of COVID-19.

## Access to community health workers, and digital divide with Rohingya and Host communities

According to our analysis, in both communities, age and health status, access to CHWS and the digital divide or the access to digital media among the Rohingya and host communities played a role in women and girls' access to accurate COVID-19 information, how they internalized it and whether they were willing to abide by safety measures against the virus.

In the Rohingya community, adolescent girls and young adults faced restrictions on mobility outside of their homes due to the rigid socio-cultural beliefs and gender norms and had little-to-no access to mobile phones, television, radio, and social media. Their main sources of COVID-19 information amid the pandemic were mostly their male family members and to some extent, the local female community youth volunteers.

Most of the younger adult Rohingya women, many of whom were pregnant and lactating, were burdened with added caregiving responsibilities during the lockdown and had no scope to talk to the CHWs. Thus, they were totally dependent on their husbands for any COVID-19 related information and were inclined towards misconceptions, rumors, and fatalistic notions about COVID-19.

Comparatively, Rohingya adolescent girls, despite being as restricted as young adults, identified CHWs as their primary sources of COVID-19 information and pointed out their trust in CHWs, how they helped them stay updated during the lockdowns (March 2020 –July 2020), and their dedication to following the safety protocols. In the words of one 17-year-old Rohingya girl,

*"I first about Corona from Jyoti (Pseudonym) Apa. When she told me that people are dying in China and America, I was so scared. . .My father will never allow me to go to the pharmacy myself if I have flu or anything. During lockdown having the flu means we would be half-dead due to fear (of contracting COVID-19) . . .Whenever Jyoti Apa came, I would listen to her. . .I would follow her suggestions. . ."* (Adolescent Girl, Rohingya).

This quote highlights the trust these girls have in local community volunteers (NGO personnel, CHWs, etc.) from their area, and notwithstanding their mobility restrictions, they were able to connect with these trusted sources, get updated about the latest COVID-19 information, and follow it, much as their situation allowed.

On the other hand, elderly women, many of whom were PWDs, were reluctant to believe in and abide by the safety measures due to their lack of mobility and access to first-hand COVID-19 information. Although their mobility issues stemmed from their weak physical health, it made them dependent on the male members of their families, similar to the young Rohingya adults.

For instance, one elderly Rohingya woman with multiple health conditions mentioned that she heard about Corona from her son, who had heard it in a gathering that he attended at a local tea stall. According to her, the virus was a rumor to frighten the elderly and people living with health conditions. In her own words,

*"My son told me about this disease (COVID-19) . . .He told me that I should not sit near the window or I may get infected. . .I think he just wants me to be on the bed all the time. . .otherwise, why would anybody fear Corona. Has anybody ever died from it?"* (Elderly Woman, Rohingya).

The situation was entirely different in the host community with most of the adolescent girls, younger women, and some middle-aged adult women having more access to information

through the internet, social networking sites, and mobile-based applications like WhatsApp, Facebook, etc. As mentioned by one adolescent girl from the host community,

> "*I heard about COVID-19 when my friend shared me the link to a newspaper article on Corona on WhatsApp. . .*" (Adolescent Girl, Host).

In contrast to the camps, most of the participants in the host sample had access to TV, radio, and social media. Many of them mentioned learning about the COVID-19 disease and its safety measures through the TV. However, their concerns stemmed from fear of the refugees whom most of them described as "*outsiders*". As a result, during the lockdowns, women and girls faced added restrictions from their families. As mentioned by one 22-year-old married woman living in one of the villages of host communities,

> "*Even though there is no rickshaw or tempo [battery-operated three-wheeler] available now-a-days [during the lockdown], my family still does not allow me to go outside because the Barmas [a local name for Rohingya refugees] are everywhere. What if they abduct me?*" (Young Woman, Host).

Compared to Rohingya older adults, the elderly participants in the host sample exhibited better knowledge about COVID-19 and its subsequent protocols, claiming better access to COVID-19 information through digital media including TV, Radio, Mobiles available in their households, etc. For instance, one adult PWD woman said that she had first heard about COVID-19 from the news on the TV and she was alarmed by it. She said, "*I was so scared that I immediately started cleaning my hands and my child's hands as well. If it's on the news then it must be very serious, right?*" *(Adult Woman Host).* Even though adolescents had more access to digital media, being in the same household indicated transmission of COVID-19 information within the family members. As was in case of Shamima (pseudonym), a 65-year-old woman living in the host communities. Shamima had attained no formal education but she could demonstrate the WHO recommended handwashing method and describe the significance of more-or-less all steps. She mentioned that her granddaughter–an adolescent girl studying in nearby community-based school, had watched a video on her phone and explained to her the importance of handwashing.

However, this level of household discussion regarding health-related issues (For e.g., COVID-19) were not referred by any of the participants in the Rohingya sample. The Rohingya participants' thoughts were mostly included towards fatalistic notions stemming from their religious beliefs, pre-existing vulnerabilities due to facing genocide, and COVID-19 seemed to be lower in the hierarchy of worries.

Interestingly, a few older adult participants in the host sample, despite access to CHWs, TV, and Radio expressed their disbelief towards the severity of COVID-19. Most of them identified religious reasons when asked about the origin of COVID-19 indicating the influence of religious values on the health beliefs. Due to this, they chose to stay confined to their rooms and prayed as they believed prayers could save them from COVID-19. This confinement, despite incidences of ailments like joint pain, headache, dental pain, etc., many older adult participants in the host sample were confined to their rooms and hardly got any chance to talk to CHWs or listen to awareness messages. Additionally, due to the unavailability of transportation and the fear of contracting COVID-19, their caregivers also denied accompanying them to the health facilities where they could have probably obtained correct information about COVID-19 that could have aided in their understanding of the severity of the disease. The religious beliefs, coupled with the fatalistic thoughts stemming from religious beliefs, and dependency on

caregivers influenced the access to COVID-19 information for older adult participants in the host community sample.

Describing the COVID-19 pandemic as "*Kiyyamat*" (the day of judgement; a religious belief in Muslim faith), and fuelled by fatalistic thoughts, one elderly woman from the host community, confined herself to her room so that she could pray. In her own words,

> "*This is Qayamat [dooms day–a concept in Muslim faith] . . .Corona will only hurt those who are disbelievers and have committed sins. . . [Besides] I have already approached my time. I cannot eat what I want, I cannot go where I want. . .Now I cannot go to hospital also. . .. [because] my son is afraid [of contracting COVID-19] in Hospital. . . I do not care about Corona. . .*" (Elderly Woman, Host).

This quote highlights how religious beliefs, dependency, and fatalistic beliefs influenced the perceptions of these women by hindering access to information.

## Socio-economic status and social support networks also determine access to information

The data found that women and girls from comparatively well-off families and/or with better social support networks in both Rohingya and host communities had more access to information.

In the Rohingya community, the socio-economic status can be depicted by the number of male members in the household who were able to work and earn income and provide for their families. In addition, working also entailed venturing outside their homes, talking to their peers, attending informal gatherings at tea stalls and prayers in the mosques. Women and girls from these households had greater access to information regarding the pandemic as the working family members (most men and boys) would learn about the pandemic from outside and share parts of that information with the women and girls at home.

For instance, one 35-year-old Rohingya woman said,

> "*One of my sons has a small shop in the Bazar and he hears many things about the world. I heard about this virus from him first. A group of customers told him and he told all of us.*" (Adult Woman, Rohingya).

Another 20-year-old young Rohingya female whose brother worked as a volunteer for an NGO had also learned about the pandemic from her brother. She mentioned,

> "*Even before the NGO workers came to us, I heard about Corona from my brother. I told all my friends as well. I often hear things from him and let my friends know.*" (Young Adult, Rohingya).

The Socio-economic status of Rohingya families was also determined by how much of the savings, jewellery and other notable good they were able to carry with them during the exodus of August 2017. Even though they had to bargain most of these items to navigate their way to the refugee camps, some families still possessed such items. These commodities held a great influence on how people perceived families which possessed them. Such families were perceived well-off by their peers which entailed that the women from the neighborhood and other active female members of the society (such as community leaders' wife, youth volunteer, etc.) would often visit them during the lockdowns and discuss COVID-19 related information

thereby aiding access to correct COVID-19 information for women and girls in such households. In the words of Morjina, a middle-aged mother of three children,

> "*We are lucky that Sameena (Pseudonym; Community Leader's wife) comes to our home. . ..
> She told me that I should be careful about the virus and ask my husband to wash thoroughly
> [when he returns from outside] . . .Romana [pseudonym, youth volunteer] and her friend also
> come to my home. . . They taught them how to tighten the mask if it is loose. . .I think they
> come to our house because we are a bit better off than others [in the block] . . .Alhamdulillah.
> We have some hard-earned items [referring to jewellery] . . .My husband had to go to the mar-
> ket to sell many times*" (Adult Woman, Rohingya).

The reputation of being "better-off" in the block put these women and their families in privilege and facilitated their access to COVID-19 information.

Similar to the participants in the Rohingya sample, participants in the host sample with economically well-off or educated families had greater access to information regarding the pandemic. This was due to the fact that often their families had TVs or devices that had access to social media or they had access to government/NGO structures that provided information. This was evident from the experience of a 25-year-old young woman whose husband owned a Mobile Phone shop. She had her own mobile phone and was able to access social media sites which is where she first learned about the pandemic. In her own words,

> "*I saw on Facebook that a virus was all over the world. I was very scared because my brother
> lives in Saudi and I did not want him to suffer alone. I asked him to come back but he didn't.
> He just asked me to stay away from everyone and wash my hands. I also saw on YouTube
> that I need to wash my hands. So now, my husband and I always wash our hands as much as
> possible.*" (Young Woman, Host).

Another 17-year-old adolescent girl whose elder brother and sister worked as NGO community volunteers also said that she had learned about the pandemic prior to any of her friends knowing what Corona was. This was because her siblings had been briefed at work and they were able to relay the information to her as well.

As was in the case of Shohana (pseudonym), a 17-year-old adolescent from host communities. Her elder brother and sister worked as volunteers in local NGOs. When COVID-19 first arrived in Bangladesh, Shohana was one of the first to know about the disease among her peers. In her own words,

> "*My sister and brother told me about it (COVID-19). At first, I did not want to believe them
> but then I saw on TV as well that it is a deadly disease. Also, they heard from their offices so it
> must have been right.*" (Adolescent Girl, Host).

## State of vulnerability

According to our analysis, an individual's state of vulnerability (pre-existing vulnerabilities before the COVID-19 pandemic) also influences their access to information regarding COVID-19.

Restricted mobility, denied citizenship, forced displacements, and enduring persecution for decades indicate the levels and extent of vulnerabilities, Rohingya women and girls live with. COVID-19 impact–loss of livelihood, food insecurity complete loss of income put such families through a strain making the women and children vulnerable to abuse especially in the contexts where husbands or male family members are the main sources of information and the

only way of linking with the outside world for the Rohingya women. Those who were victims of intimate partner violence (IPV) were often deprived of getting information from that only source.

As was in the case of one 27-year-old Rohingya woman who faced intimate partner violence (IPV) and believed that she did not possess the means to be worried about a virus. She was not at liberty to worry about COVID-19 because asking questions about it would trigger an episode of IPV in her household. In terms of risks, COVID-19 appeared low in the hierarchy of concerns, and she was overcome with the daily fears of physical beatings and abuse from her husband. According to her,

> "*How will I worry about this disease when anytime I ask my husband about it, he gets angry? I don't want to anger him and create problems for myself. I don't even know what it is, so why should I worry?*" (Adult Woman, Rohingya).

Conversely, the host community also lived in a vulnerable state but evidently, their vulnerability was of a different kind stemming from poverty as they lived in one of the most impoverished sub-districts of Bangladesh. Although participants in the host community sample, generally expressed having greater mobility than those in the Rohingya community and thus more access to COVID-19 information. However, for some participants, particularly the single female household heads, access to COVID-19 information was challenging. The socio-cultural and gender norms restricted their mobility and hindered their access to the COVID-19 information, and for those who had mobility of some sort, the geographical barriers–hilly terrains, muddy roadways, and lack of transportation due to COVID-19 -induced lockdown in the area made it difficult for them to access the health facilities where they could probably obtain the required COVID-19 information.

One 32-year-old single woman from the host community said, "*I wish I could earn money. But I can't. Since I don't go out, I don't know about this virus. But I've heard from my children that some virus is here. We can barely eat. . .virus is far away from my mind.*" (Single Woman, Host).

This quote highlights the state and extent of the pre-existing vulnerabilities of single women living in host communities which limited their access to COVID-19 information.

## Power and privilege

According to our analysis, power within a community and the associated privileges exist even in the direst contexts including the stateless Rohingya community and the impoverished Bangladeshi host community. Within the women and girls in the Rohingya and host communities, power and privilege exist primarily in two ways; one is direct power which leads to privilege such as a woman being a CHW or an NGO volunteer. The other is indirect power which also leads to privilege such as being the wife or daughter of a community leader or from a well-connected socio-economically better off family. Therefore, a female 22-year-old Rohingya NGO volunteer was far more aware and well-networked about the pandemic and had access to support from NGOs and other stakeholders than other Rohingya women her age. She shared,

> "*I wear masks and always ask others to wear them as well. I know this is how we can be safe from COVID-19. I hope my work has helped others to gain some knowledge about this disease.*" (NGO Volunteer, Rohingya).

On the other hand, the 45-year-old wife of a "*Majhi*" (community leader) also had more access to information and resources regarding the virus as her husband was directly involved with the NGOs that were providing information and other resources. She said,

"*My husband came and told our family about Corona when he heard from the NGO people. I was the first one to tell other women in my block. I also told them to wash their hands and not sit so close to one another when we sit and talk. I also told them to stop their children from going out and playing.*" (Adult Woman, Rohingya).

Similarly, a 15-year-old adolescent girl who belonged to a family that was economically better off, in the host community, had greater access to information. She said that her family had not suffered financially due to the lockdown and she had been able to continue her studies as she had access to the internet. According to her,

"*I study online. I also heard about this (COVID-19) from Facebook first. It came from China, right? I told my mom to wash everything when I heard about it coming to Bangladesh.*" (Adolescent Girl, Host)

However, this was not the case for all adolescent girls. Many adolescent girls had to discontinue their studies due to the pandemic and the closure of educational institutions. While many of them were married off and others suffered from the pressure of a looming early marriage. As was in the case of Aruna Khatun (pseudonym), a 16-year-old adolescent girl, who had to drop out of school as the lockdown ensued. Her father was a day laborer who lost his income as soon as the government imposed a lockdown and their family's situation was not such that she could join online classes. Aruna had to stay at home and do household chores and take care of her younger siblings and she lost her social connections and networks from where she could get information because of her inability to attend online classes. In her own words,

"*I don't know if I will be able to study again. I wanted to be a nurse. To help people. . .. now I just hope I don't have to get married soon. My parents have already talked about it a bit. They are looking for a boy. . .but I don't want to get married.*" (Adolescent Girl, Host).

## Physical (dis)ability

According to our analysis, in the Rohingya and host communities of Cox's Bazar, physical ability or disability also dictates the access to information of the women and girls regarding COVID-19. The women and girls with disabilities in these communities are not only dependent on others for their mobility but also for their information regarding the pandemic. In the Rohingya community, this is further prevalent as the terrain is hilly (and vulnerable to landslides), and a PWD would have a difficult time navigating it by themselves thus hindering their mobility. Coupled with the strict socio-cultural and gender norms, women and girls with disabilities have even less access to information. An 18-year-old Rohingya female with a disability said,

"*I was unable to attend the meetings held by NGOs because of my sickness (disability) but I heard from others that people with Corona symptoms should visit hospitals. But many people don't.*" (PWD, Rohingya).

She further added that she did not know much about the virus because she could not venture outside and the little, she knew was all from her father and brother.

In the host community, many PWD participants pointed out that they were dependent on their caregivers for mobility and information. As COVID-19 lockdown was enforced in March

2020, their caregivers could not go out and were disconnected from their peers, friends, relatives, etc. As a result, the caregivers had little-to-no information about COVID-19 and so did the PWD participants. As was in the case of Uzma, a young, 25-year-old PWD woman in the host community. She did not express much anxiety about COVID-19. She was dependent on her caregiver for all the information relating to the pandemic and she said, *"I don't really know much. But that's okay. Because she (caregiver) knows, and if something bad happens she is supposed to tell me. How else can I know if she does not tell me?" (PWD, Host).*

## Discussion

The findings of this study elucidate that the many intersectional factors have played an integral role in determining their access to information of women and girls in the Rohingya and host communities of Cox's Bazar regarding the COVID-19 pandemic. Fig 2 presents a visual framework that depicts the interplay of intersectional factors that influenced the access to information of women and girls in the studied contexts.

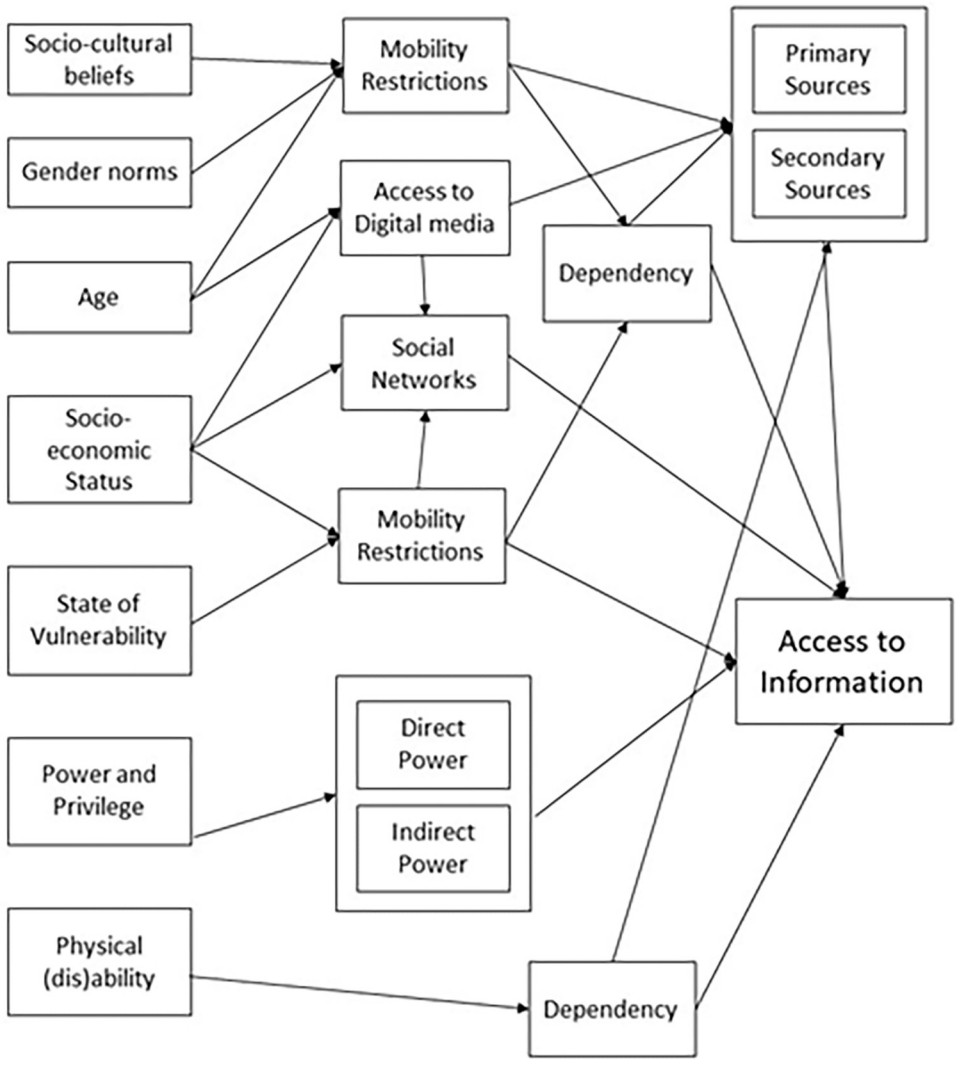

**Fig 2. Visual framework that depicts the intersectional interplay of factors that influenced the access to information of women and girls during the COVID-19 pandemic.**

First, gender played an important role in how the women and girls across age groups in both communities gathered and accessed information because the rigid socio-cultural norms stemming from deep-rooted orthodox religious beliefs perpetuated certain gender norms in the communities leading to mobility restrictions and resulting in dependency on secondary sources that can be misleading and confusing as it is often the partial information.

Women and girls, in many other humanitarian contexts [37–39], have limited access to the world outside their homes and thus have less access to information regarding the COVID-19 pandemic and how to maintain safety measures [4]. The younger age group in both communities had more access to primary sources of information and wider social support networks, while the older age groups were more dependent on secondary sources and their social networks were limited to their family members due to their difficulties in movement stemming from age-related complications. This meant that for these women and girls, information was often delayed or not relayed properly leading to misinformation and gaps in information. Additionally, because women and girls had to depend on their male counterparts for information this may have increased their dependency on men which could lead to increased cases of IPV and other forms of gender-based violence (GBV). Many situation analyses conducted during the COVID-19 by the humanitarian service providers in other humanitarian contexts also highlight similar findings [40–43] i.e., an increase in IPV and domestic violence cases among women and girls during COVID-29 lockdown [24]. This is consistent with the global data and highlights the heightened risk these women and girls face, and that COVID-19 has exacerbated these vulnerabilities [3, 5, 7, 13, 21, 26].

The socio-economic status of women and girls in these communities also affected their access to information. Our evidence suggests that in the Rohingya community, a better socio-economic status meant that the household had more earning members, mostly men. This also means that women and girls were more dependent on them for information regarding the pandemic. Conversely, socioeconomic status also indicated a privilege, and such families had better access to COVID-19 information as more socially active members like youth volunteers, community health workers, etc., visited them time and again. We thus highlight that women and girls who work as community health workers or volunteers have more access to information and are able to create awareness amongst their social circles. As such, their power and privilege aid those who are near to them. However, due to the rigid gender norms prevalent in society, this power and privilege can have added risks for women and girls. The community people may believe that they are changing gender norms through their work which may create issues for them. Additionally, if these working women happen to be younger, that may add another layer of risk as their families may want them to stop working and get married early [20]. Evidence suggests that child marriage is used as a coping mechanism for the poorest host community households and is known to increase aftershocks such as natural disasters. For instance, after the influx of Rohingya refugees, a survey reported that 23% of host communities reported an increase in child marriage practices within three months, indicating an increase in the use of this practice as a negative coping mechanism [28]. It is to be noted that the increased strain on resources due to the COVID-19 pandemic may boost the use of negative coping mechanisms as is evident from increasing GBV, IPV, and child marriage cases in both communities [11].

The state of vulnerability or pre-existing vulnerabilities of women and girls in both communities directly impacted their access to information. In the Rohingya community, women and girls faced greater vulnerabilities in the camps with increased rates of IPV and GBV [20]. The IOM's Needs and Population Monitoring (NPM) report estimates that 12 percent of households in the Rohingya community are likely to be female-headed and that 17.35 percent of the Rohingya mothers are single mothers [11]. In both communities, single female household

heads were in a uniquely vulnerable position where they were confined to their households due to rigid socio-cultural and religious norms. This directly impacted their access to information as they remained confined in their homes and also compounded the mental distress they faced amid a global pandemic. The extent of strain COVID-19 has put the single female-headed household through, it is likely that children from such households may engage in child labor through informal economic activities or drug trafficking, both of which are highly prevalent in Ukhiya, Cox's Bazar [11]. As of January 2018, over 85 percent of children in host communities were predicted to be engaged in paid and unpaid labor, one of the highest percentages relative to other groups studied (Rohingya in camps, Rohingya in settlements and Rohingya in host communities) [32].

PWDs were particularly affected due to lack of information as they often did not have the ability to collect the information by themselves. Additionally, their dependency on caregivers also limited their access to information and COVID-19 services altogether. This is in line with one report on the Asia Pacific region where women with disabilities detailed their lack of understanding regarding the pandemic due to a lack of access to information about it [44]. There is compelling evidence that PWDs and older persons face significant barriers to accessing vital services such as latrines, water stations, health centers, and assistance distribution due to mountainous and flood-prone terrain and a lack of adapted facilities and inclusive initiatives [32]. Social stigma is a major barrier preventing access to essential services and participation in community activities and decision-making processes [28]. Specialized support is restricted for persons seeking mental health and psychiatric treatment, and there are few certified experts available to meet the scope of these demands in the camps [15].

## Conclusion

This paper explored the interplay of factors that affect the access to COVID-19 information of women and girls in the Rohingya and host communities of Cox's Bazar amid the COVID-19 pandemic through an intersectionality lens. The findings revealed that both Rohingya and Bangladeshi women and girls face many barriers to accessing COVID-19 information in these communities because of their gender, socio-cultural beliefs, age, socio-economic status, state of vulnerability, power, and privileges, and the prevalent patriarchal norms which suppress their rights.

In any humanitarian crisis where communities are already facing multifaceted risks, it is imperative that services geared towards them are contextualized and keep in mind the myriad barriers to access for the communities. While planning COVID-19 interventions, intersectionality should be used as an approach to recognize critical aspects of the populations which if ignored may produce pockets of vulnerable populations within the society [9, 10]. This implies first identifying the axes of intersectionality and then considering them while planning responses. Any precautionary measure to COVID-19 that is disseminated in Cox's Bazar must be mindful of the intersectional identities of women and girls and the barriers they face to access services and information because of their intersectional identities.

## Acknowledgments

The authors are also thankful to the larger research team members; Professor M. Shafiq Rahman, Kazi Sameen Nasar, Saifa Raz, Abdul Jabbar Topu, ASM Nadim, Zuhrat Mahfuza Inam, and Dr. Ashrafuzzaman Khan, for their support during the data collection.

## Author Contributions

**Conceptualization:** Ateeb Ahmad Parray, Muhammad Riaz Hossain, Rafia Sultana, Bachera Aktar, Sabina Faiz Rashid.

**Data curation:** Ateeb Ahmad Parray, Muhammad Riaz Hossain, Rafia Sultana.

**Formal analysis:** Ateeb Ahmad Parray, Muhammad Riaz Hossain, Rafia Sultana, Bachera Aktar, Sabina Faiz Rashid.

**Funding acquisition:** Sabina Faiz Rashid.

**Investigation:** Ateeb Ahmad Parray, Muhammad Riaz Hossain, Rafia Sultana, Bachera Aktar.

**Methodology:** Ateeb Ahmad Parray, Muhammad Riaz Hossain, Rafia Sultana, Bachera Aktar, Sabina Faiz Rashid.

**Project administration:** Ateeb Ahmad Parray, Muhammad Riaz Hossain, Rafia Sultana, Bachera Aktar, Sabina Faiz Rashid.

**Resources:** Ateeb Ahmad Parray, Muhammad Riaz Hossain, Rafia Sultana, Bachera Aktar, Sabina Faiz Rashid.

**Software:** Muhammad Riaz Hossain, Rafia Sultana.

**Supervision:** Ateeb Ahmad Parray, Muhammad Riaz Hossain, Bachera Aktar, Sabina Faiz Rashid.

**Validation:** Ateeb Ahmad Parray, Muhammad Riaz Hossain, Bachera Aktar.

**Visualization:** Ateeb Ahmad Parray, Muhammad Riaz Hossain, Rafia Sultana.

**Writing – original draft:** Ateeb Ahmad Parray.

**Writing – review & editing:** Ateeb Ahmad Parray, Muhammad Riaz Hossain, Rafia Sultana, Bachera Aktar, Sabina Faiz Rashid.

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
