## [Decision Letter · Decision Letter 0]

4 Jul 2022

PGPH-D-22-00607

“Younger women had more access to COVID-19 information”: An Intersectional Analysis of Factors Influencing Women and Girls' Access to COVID-19 Information in Rohingya and Host Communities in Bangladesh

Dear Mr. Ateeb Perray,

Thank you for submitting your manuscript to PLOS Global Public Health. After careful consideration, we feel that it has merit but does not fully meet PLOS Global Public Health’s publication criteria as it currently stands. Therefore, we invite you to submit a revised version of the manuscript that addresses the points raised during the review process.

We look forward to receiving your revised manuscript.

Kind regards,

Zahra Zeinali, MD MPH

Academic Editor

Journal Requirements:

1. Please provide additional details regarding participant consent. In the ethics statement in the Methods and online submission information, please ensure that you have specified whether: 1) whether the ethics committee approved the verbal/oral consent procedure, 2) why written consent could not be obtained, and 3) how verbal/oral consent was recorded. If your study included minors, please state whether you obtained consent from parents or guardians in these cases. If the need for consent was waived by the ethics committee, please include this information.

2. Please update the Funding Information in the system to matched with the Financial Disclosure Statement.

3. "Please provide separate figure files in .tif or .eps format and removed from the manuscript file.

4. Please update the 'Competing Interests' statement with this "The authors have declared that no competing interests exist".

Additional Editor Comments (if provided):

Dear Mr. Ahmad Parray, 

  Thank you for submitting your manuscript to PLOS Global Public Health. After careful reviews, we are happy to inform you that the manuscript can be ready for publication granted some minor revisions and improvements are applied. Therefore, we invite you to submit a revised version of the manuscript that addresses the points raised during the review process. Please review the comments from reviewers and address them in the revised version. 

Thank you.

Reviewers' comments:

Reviewer's Responses to Questions

**Comments to the Author**

1. Does this manuscript meet PLOS Global Public Health’s publication criteria? Is the manuscript technically sound, and do the data support the conclusions? The manuscript must describe methodologically and ethically rigorous research with conclusions that are appropriately drawn based on the data presented.

Reviewer #1: Yes

Reviewer #2: Yes

2. Has the statistical analysis been performed appropriately and rigorously?

Reviewer #1: No

Reviewer #2: N/A

3. Have the authors made all data underlying the findings in their manuscript fully available (please refer to the Data Availability Statement at the start of the manuscript PDF file)?

Reviewer #1: Yes

Reviewer #2: Yes

4. Is the manuscript presented in an intelligible fashion and written in standard English?

Reviewer #1: Yes

Reviewer #2: Yes

5. Review Comments to the Author

Reviewer #1: The introduction needs to be expanded - while it makes it clear what will be discussed in the paper, it falls short of providing a sufficient overview before delving into the topic. For readers that are entirely unaware of the context, a bit more needs to be presented there. The introduction is also recommended to conclude with the main research question. Although this is clear in the title, it still needs to be clearly reformulated in the text).

Reviewer #2: This is a well articulated paper, which can add a real value in understanding the context of COVID-19 among Rohingya and Host community women from an intersectionality perspective. I have just a few observations and some minor suggestions.

1. Explanation of purda (line 132 and line 313) has been repeated twice. Authors can remove the repetition.

2. The findings presented from line 436 to the end of that section match better with socio-cultural belief section. Because those lines discussed the fatalistic religious belief hampered access to information.

3. Some points discussed in power and privilege section, (from line 569) in my opinion, would better match in the socio-economic background section, particularly those cases where school dropout and early marriage happened due to job loss of parents. In fact, power and privilege section does not properly reflect that people like NGO workers are holding ‘power’ for which they are getting extra advantage. They are in ‘privileged’ position though, as findings suggest. The authors may rethink about the word ‘power’ here.

4. It looks like a word is missing in line 365

5. Some typos need to be checked.

6. PLOS authors have the option to publish the peer review history of their article (what does this mean?). If published, this will include your full peer review and any attached files.

**Do you want your identity to be public for this peer review?** For information about this choice, including consent withdrawal, please see our Privacy Policy.

Reviewer #1: **Yes: **Jasmin Lilian Diab

Reviewer #2: No

---

## [Decision Letter · Decision Letter 1]

3 Nov 2022

“Younger women had more access to COVID-19 information”: An Intersectional Analysis of Factors Influencing Women and Girls' Access to COVID-19 Information in Rohingya and Host Communities in Bangladesh

PGPH-D-22-00607R1

Dear Mr. Parray,

We are pleased to inform you that your manuscript '“Younger women had more access to COVID-19 information”: An Intersectional Analysis of Factors Influencing Women and Girls' Access to COVID-19 Information in Rohingya and Host Communities in Bangladesh' has been provisionally accepted for publication in PLOS Global Public Health.

Best regards,

Julia Robinson

Executive Editor

Reviewer Comments (if any, and for reference):

Reviewer's Responses to Questions

**Comments to the Author**

1. If the authors have adequately addressed your comments raised in a previous round of review and you feel that this manuscript is now acceptable for publication, you may indicate that here to bypass the “Comments to the Author” section, enter your conflict of interest statement in the “Confidential to Editor” section, and submit your "Accept" recommendation.

Reviewer #1: All comments have been addressed

Reviewer #2: All comments have been addressed

2. Does this manuscript meet PLOS Global Public Health’s publication criteria? Is the manuscript technically sound, and do the data support the conclusions? The manuscript must describe methodologically and ethically rigorous research with conclusions that are appropriately drawn based on the data presented.

Reviewer #1: Partly

Reviewer #2: Partly

3. Has the statistical analysis been performed appropriately and rigorously?

Reviewer #1: N/A

Reviewer #2: I don't know

4. Have the authors made all data underlying the findings in their manuscript fully available (please refer to the Data Availability Statement at the start of the manuscript PDF file)?

Reviewer #1: Yes

Reviewer #2: Yes

5. Is the manuscript presented in an intelligible fashion and written in standard English?

Reviewer #1: Yes

Reviewer #2: No

6. Review Comments to the Author

Reviewer #1: (No Response)

Reviewer #2: 1. In page 13 author/s mentioned favourable/ unfavourable attitude. What does it mean?

2. Methodology and data analysis looks alright. However, the discussion part is poorly structured. There are numerous mistakes in sentence structure, which make the discussion confusing. The author/s brought some study reference and mentioned whether their findings match with the referred articles or not. I found this not a proper academic way. If the findings have to be cross checked with other relevant literature, the authors should be careful to relate and contextualize. It should be a more in depth and critical discussion.

3. In conclusion, ‘the attitude of women about partner’s involvement was adequate” .. this is not clear. How can an attitude become adequate?

4. The male involvement is only seen whether men is supporting women or accompanying in health center. The active involvement of men as contraceptive users should be included as an important critical point in the introduction, analysis and in the discussion section.

5. the whole article needs a thorough revision for grammar check. i suggest the authors take assistance from a language editor .

7. PLOS authors have the option to publish the peer review history of their article (what does this mean?). If published, this will include your full peer review and any attached files.

**Do you want your identity to be public for this peer review?** For information about this choice, including consent withdrawal, please see our Privacy Policy.

Reviewer #1: **Yes: **Jasmin Lilian Diab

Reviewer #2: **Yes: **Sanzida Akhter
